

# The role of droplet sedimentation in the evolution of low level clouds over southern West Africa

Christopher Dearden[1,a], Adrian Hill[2], Hugh Coe[1], and Tom Choularton[1]

[1]Centre for Atmospheric Science, School of Earth and Environmental Science, University of Manchester, United Kingdom
[2]Met Office, Exeter, United Kingdom
[a]now at: Centre of Excellence for Modelling the Atmosphere and Climate, School of Earth and Environment, University of Leeds, United Kingdom

*Correspondence to:* Christopher Dearden (c.dearden@leeds.ac.uk)

**Abstract.** Large eddy simulations are performed to investigate the influence of cloud microphysics on the evolution of low level clouds that form over southern West Africa during the monsoon season. We find that, even in clouds that are not precipitating, the size of cloud droplets has a non-negligible effect on liquid water path. This is explained through the effects of droplet sedimentation, which acts to remove liquid water from the entrainment zone close to cloud top, increasing liquid water path.

Sedimentation also produces a more heterogeneous cloud structure and lowers cloud base height. Our results imply that an appropriate parameterization of the effects of sedimentation is required to improve the representation of the diurnal cycle of the atmospheric boundary layer over southern West Africa in large-scale models.

*Copyright statement.* TEXT

## 1   Introduction

During the months of June to September, the climate of southern West Africa (SWA) is dominated by the southwesterly flow of the West African Monsoon (WAM), which is principally driven by a north-south pressure gradient associated with the Saharan heat low and brings seasonal rains to the region (e.g. Sultan and Janicot 2000, LeBarbé et al. 2002). Clouds, through their diabatic effects, are known to exert an influence on the WAM circulation. For example, a number of studies have explored the role of moist convection in the Sahel (e.g. Garcia-Carreras et al. 2013; Marsham et al. 2013; Birch et al. 2014), revealing that

the diurnal cycle of latent heating and cloud radiative forcing affect the north-south pressure gradient and hence the northward advection of moisture.

Low level clouds (LLCs) over SWA, with bases only a few hundred metres above ground level (agl), are also a common occurrence during the WAM season (e.g. Schrage and Fink 2012; van der Linden et al. 2015), yet it is only recently that their role has been considered in detail. LLCs typically form near the Guinea Coast sometime after sunset following the initiation of

the southwesterly nocturnal low level jet. The jet is linked to the low-level pressure gradient, supplying moisture to the Sahel region where it is mixed as a result of convection during the day (Parker et al., 2005; Lothon et al., 2008; Abdou et al., 2010;



Bain et al., 2010). The clouds then spread northwards inland during the night (Schuster et al., 2013; van der Linden et al., 2015; Kalthoff et al., 2017), and typically persist until the late morning after which they transition to broken cumulus and dissipate. Through their impact on surface solar irradiance, the LLCs play an important role in the evolution of the atmospheric boundary layer (Gounou et al., 2012), and the regional climate of West Africa (e.g. Knippertz et al. 2011; Hannak et al. 2017).

5    The most comprehensive observational study of the atmospheric boundary layer over SWA was conducted recently by Kalthoff et al. (2017) during the DACCIWA field campaign (Knippertz et al., 2015a, 2017; Flamant et al., 2017). Between 14 June and 30 July 2016, intensive observations were made at three ground-sites - Savé (Benin), Kumasi (Ghana) and Ile-Ife (Nigeria) - using a variety of instrumentation including radiosondes and wind profilers (Derrien et al., 2016), radars and ceilometers (Handwerker et al., 2016) and microwave radiometers (Wieser et al., 2016). These ground-based observations were complemented by in-situ measurements of aerosol and cloud properties from three European aircraft, which together conducted 50 research flights between 27 June and 16 July. The results presented in Kalthoff et al. (2017) reveal significant variability in the onset and dissolution of LLCs over southern West Africa from day-to-day and from site-to-site. However the governing processes and mechanisms responsible are not fully understood. Furthermore, large-scale models struggle to represent these LLCs and their variability accurately. Hannak et al. (2017) found that many current GCMs suffer a common bias in the form of insufficient low cloud cover over SWA, abundant solar radiation, and thus too large a diurnal cycle in temperature and relative humidity. They concluded that targeted model sensitivity experiments are needed to test possible feedback mechanisms between low clouds, radiation, boundary layer dynamics, precipitation, and the WAM circulation.

Several studies have proposed specific mechanisms relevant for the formation and break up of the cloud decks (e.g. Schrage and Fink 2012; Schuster et al. 2013; Adler et al. 2017). Specifically, LLCs are believed to be sensitive to temperature and moisture advection from the south (controlled by the strength of the low level jet), vertical mixing of heat and moisture arising due to shear-generated turbulence, radiative cooling at cloud top, condensational heating, sub-cloud evaporation, orographic lifting and lifting induced by gravity wave propagation. In addition to each of these processes, it is important to consider also the role of microphysics in the evolution of LLCs, and the potential modification of the cloud properties via the interaction with aerosols (Knippertz et al., 2015a, b). The combination of ground-based and in-situ measurements from DACCIWA offer a unique opportunity to explore the links between aerosols, microphysics and the bulk cloud properties in SWA, and to inform an appropriate level of parameterization for the representation of LLCs in regional and global models.

The purpose of the present study is to address this issue by performing large eddy simulations of a selected DACCIWA case study, with a focus on testing the ability of the model to reproduce the observations through a consideration of the treatment of the cloud microphysics. In doing so, the aim is to identify an optimum configuration for the parameterization of boundary layer clouds over SWA. The rest of this paper is organised as follows. Section 2 presents details of the case study to be simulated; section 3 describes the numerical model used to perform the simulations, along with details regarding model configuration and initialisation, and the results are presented in section 4. Implications of the findings are discussed in section 5, before the main conclusions are summarised in section 6.



## 2  Case study

For the purpose of this study, we focus on a particular case on 4th-5th July 2016, the 7th Intensive Observation Period (IOP) from the DACCIWA field campaign (Flamant et al., 2017). The conditions on this day were fairly typical of the campaign as a whole, with observations collected at the ground site at Savé revealing the onset of the low level jet around 1800 UTC

on the 4th July, followed by the formation of a low-level stratocumulus deck during the night. Cloud at the Savé ground site was first observed at 0300 UTC on the 5th July, which persisted until around mid-day local time after which it began to break up into patchy cumulus (see Flamant et al. 2017, their Fig.6). No precipitation was recorded at the Savé ground site for this case, consistent with the majority of days sampled during the campaign period. For an overview of the diurnal cycle of the atmospheric boundary layer at Savé during DACCIWA, the reader is referred to Kalthoff et al. (2017).

The radiosonde data from IOP 7 provide more information on the structure and evolution of the boundary layer on this day. Profiles of potential temperature and relative humidity from the 0330 UTC sonde, launched approximately half an hour after the cloud was first detected at Savé, are shown in Fig 1a-b. The relative humidity profile reveals a cloud layer approximately 200m thick, with a cloud top height of  550m capped by a temperature inversion of 1.5 K. The horizontal wind components, shown separately in Fig 1c-d, reveal a low level jet with a wind speed maximum at a height of 550m agl, and the cloud layer

located directly beneath. Later sondes from 0500, 0628, 0800 and 0928 UTC (not shown) reveal that the cloud layer persisted throughout the morning, with the relative humidity occasionally peaking just below water saturation, suggesting the presence of some breaks in the cloud cover. This is consistent with images from the infrared camera at Savé (see Fig. 2). The low level jet persisted until around 1100 UTC (Fig. 3), by which time the depth of the boundary layer had increased to 1 km due to solar heating of the surface, resulting in lifting of the cloud deck (as shown in Fig 1a-b). Three research aircraft were also deployed

in sequence on this day, taking in-situ measurements along the transect between Lome and Savé from 0800 UTC through to 1800 UTC in order to sample the microphysical evolution during the cloud lifecycle (Flamant et al., 2017).

## 3  Model description

To fulfill the needs of this study we utilise the Met Office/NERC Cloud model (MONC; Brown et al. 2015). MONC is a rewrite of the original Met Office Large Eddy Model (LEM), which has been used extensively over the past twenty years to study

cloud processes in a variety of regimes (e.g.  Brown 1999, Brown et al. 2002, Clark et al. 2005, Connolly et al. 2006, Marsham et al. 2006, Connolly et al. 2013, Young et al. 2017). MONC offers several key advantages over the original LEM, including code optimisations, bug-fixes and a new solver that enables simulations to be performed with relatively large domain sizes without having to compromise on the model resolution.

Radiation is represented in MONC by the Suite of Community Radiative Transfer codes based on Edwards and Slingo

(SOCRATES; Edwards and Slingo 1996), the same as that used in the Met Office Unified Model, specifically the Global Atmosphere Model 6.0 (Walters et al., 2017). SOCRATES is called on a three minute timestep, allowing the effects of long wave cloud top cooling and short wave absorption within the cloud layer to be captured in the model.





Regarding the treatment of cloud processes, MONC can be coupled to the CASIM (Cloud-AeroSol-Interacting-Microphysics) module, a newly developed user configurable multi-moment scheme that represents five hydrometeor species (cloud, rain, ice, snow and graupel) and multi-mode aerosols. CASIM has already been used within the Met Office Unified Model to study aerosol-cloud interactions in different meteorological contexts, e.g. Grosvenor et al. (2017), Miltenberger et al. (2017)
and Stevens et al. (2017). There is also the option to run MONC using a comparatively basic 'all or nothing' cloud scheme that represents only the effects of condensation and evaporation using a single prognostic variable (the cloud mass mixing ratio). In this scheme, there is no sedimentation of cloud or rain, no autoconversion or accretion, and no aerosol information specified. In the present study, MONC experiments are performed using both CASIM and the simple cloud scheme, with further details given in section 3.2.

## 3.1 Model initialisation and configuration

MONC is initialised using profiles of potential temperature, total water mass mixing ratio and horizontal wind components, which are obtained from radiosondes launched from the Savé ground site. For IOP 7, we initialise the model using data from the 0330 UTC radiosonde as shown in Fig. 1, interpolating the data onto the model grid with a vertical resolution of 10 m. Where the initial relative humidity profile is at water saturation (i.e. between 350 m and 550 m in Fig. 1b), the cloud liquid
water mass mixing ratio profile is calculated assuming an adiabatic cloud parcel ascent from cloud base to cloud top. The profile of total water mass mixing ratio is then calculated as the sum of the cloud liquid water and water vapour mass mixing ratios at each model level. During the first model timestep, this supersaturated profile results in the immediate production of a cloud layer via condensation, and at an early enough stage in its lifecycle to study its subsequent evolution over a period of 7.5 hours. The choice of the 0330 UTC sonde for initialisation is justified since the aim of the present study is to focus on the role
of microphysical factors that control the subsequent evolution of the LLC, rather than the meteorological factors that govern the onset of cloud formation.

Regarding the forcing of the wind field, the winds from 0330 UTC are relaxed towards the u and v wind components from the 1100 UTC radiosonde (as shown in Fig. 1c-d) over a period of 7.5 hours. This allows the model to maintain the low level jet throughout the simulation period. No forcing increments are applied to either the potential temperature field or the moisture
fields; however a constant large-scale divergence of 5.e-6 s$^{-1}$ is imposed throughout the model domain, to produce a constant large-scale subsidence. According to the ERA-Interim reanalysis dataset (Dee et al., 2011), this value lies within the variability range over southern West Africa during the time period of the DACCIWA field campaign.

Importantly, MONC is not coupled to an interactive land surface scheme in the present study and so to represent the effects of the surface, time-varying fluxes of sensible and latent heat are prescribed using surface measurements from the Savé ground-
30 site (Kohler et al., 2016). Fluxes from 5th July 2016 used to force the model are plotted in Fig. 4, for the simulation period indicated.

All the simulations presented in this paper use a domain size of 7.5 km x 7.5 km in the horizontal with a 30 m grid-spacing, and a vertical extent of 2 km with a 10 m spacing between vertical levels up to 1.5 km, increasing to a 20 m spacing between 1.5 km and 2 km. The top 500 m is a damping layer to prevent unwanted gravity waves from reflecting off the rigid model




lid. The first two hours of each simulation (between 0330 - 0530 UTC) are discarded to allow for model spin up, and periodic boundary conditions are used in all cases.

## 3.2 Details of model experiments

Here we introduce and describe two initial experiments, the results from which are analysed in the next section.

The first MONC experiment makes use of CASIM, and is configured for dual moment cloud and rain, while cold processes were not considered or required. Autoconversion and accretion are represented using the scheme of Khairoutdinov and Kogan (2000), and sedimentation of cloud droplets and rain is included. A saturation adjustment scheme is employed for condensation and evaporation of cloud droplets, while rain evaporation is based on the scheme used in the LEM (Gray et al., 2001). CASIM includes various options for aerosol activation and in the this work we employed the scheme of Abdul-Razzak et al.

(1998), with the aerosol specified as a single accumulation mode log-normal size distribution following the analysis of regional aerosol properties in Haslett et al. (in preparation, 2018). The aerosol mass and number fields are completely passive in this experiment (i.e. not influenced by cloud and rain processes), and are used only to determine the number of droplets activated. This experiment is henceforth referred to as CASIM_NO_PROC.

The second MONC experiment replaces CASIM with the 'all or nothing' cloud scheme; we refer to this experiment as

SIMPLE_CLOUD. Since no precipitation was observed during the majority of the DACCIWA IOPs, including the present one, the rationale of this second experiment is to explore whether a simulation with such a basic representation of cloud is able to reproduce the observations for this case, and therefore to reveal the extent to which the additional complexity offered by CASIM impacts the simulation.

## 4    Results

We begin with an initial inspection of results from the CASIM_NO_PROC experiment. Fig. 5 shows a time-height plot of the domain average cloud mass mixing ratio for the period 0530 UTC – 1100 UTC. The presence of a cloud layer is revealed with an initial mean cloud base around 350 m, and a cloud top of 600 m. Following sunrise at 0537 UTC, the surface fluxes of sensible and latent heat increase sharply from around 0700 UTC as shown in Fig 4, resulting in a deeper boundary layer (BL) and lifting of the cloud layer from around 0800 UTC. The general trend in the timeseries of cloud base height is well captured

by the model, as seen in the comparison against the ceilometer measurements from the Savé ground-site (fig 6). Cloud top long wave radiative cooling was found to be crucial for the development and maintenance of the cloud, through the generation of an overturning circulation within the cloud layer. Indeed, without any long wave cooling, the model was unable to sustain the cloud layer, resulting in complete dissipation by the end of the spin-up period.

Figure 7 provides further information about the evolution of the mixing state of the simulated BL, in terms of profiles of

liquid water potential temperature, liquid water mixing ratio and total water mixing ratio following the diagnostic analysis of Jones et al. (2011). Domain average profiles from 0530 UTC (fig 7a) reveal a predominantly well-mixed cloud-topped BL capped by a temperature inversion at 600m agl. A stable layer exists from the surface up to 150m agl, consistent with long





wave cooling of the surface during the night, with a thin fog layer which dissipates by 0630 UTC. By 1100 UTC (fig 7b), the increase in surface fluxes produces a deeper, convective BL with an unstable layer at the surface. A well-mixed layer exists between 50m and 400m agl in the sub-cloud region, with a hint of a second shallower well-mixed layer directly below the top of the BL, where the values of liquid water mixing ratio are largest. These layers are separated by a transition region between

400m and 900m agl, where the liquid water potential temperature gradually increases with height. The model captures the deepening of the BL as seen in the observations, with a simulated BL height of 1.1 km by 1100 UTC compared with  1 km in the corresponding radiosonde profile (Fig 1a,b). The vertical structure of both potential temperature and relative humidity are also well captured by the model, as shown in (Fig 8a,b).

Fig. 9 compares the timeseries of simulated LWP from CASIM_NO_PROC with observations from the vertically pointing

ground-based microwave radiometer at Savé (Wieser et al., 2016). Because the radiometer measurements represent the time evolution at a single location, care must be taken when evaluating the model against this dataset to account for the difference in spatial sampling. Hence in Fig. 9 we plot both the simulated LWP timeseries taken from the centre of the model domain diagnosed at 1 minute intervals, together with the variability in LWP across the whole domain. The model simulates the evolution of LWP in a manner that is broadly consistent with the measurements, with the observations for the most part lying

within +/- 2 standard deviations of the simulated LWP values. Peak local values of LWP also occur at approximately the correct time in the model as well, i.e. after 0800 UTC when the surface fluxes have started to rise sharply. No precipitation was produced by the model during the simulation period, consistent with the measurements at Savé.

Having validated the ability of CASIM_NO_PROC to capture the key features of the observations, we now consider the impact of reducing the complexity of the cloud scheme by analysing results from the SIMPLE_CLOUD experiment. Fig 10

shows that SIMPLE_CLOUD underestimates the variability in LWP before 0730 UTC compared to both the observations and CASIM_NO_PROC. Maps comparing the spatial distribution of LWP within the model domain for both simulations (Fig. 11) confirm that the cloud is much more spatially homogeneous in SIMPLE_CLOUD initially, resembling a largely featureless sheet of stratus as opposed to the more lumpy stratocumulus seen in CASIM_NO_PROC. A comparison of the timeseries of mean LWP in the domain (Fig 12) reveals that, although both simulations show a similar rise and fall pattern with a

peak around mid-morning, there are still some notable differences despite neither simulation producing any precipitation. For instance, following completion of the spin-up phase, the values of LWP are very similar. But between 0530 - 0700 UTC, the rate of LWP growth slows in SIMPLE_CLOUD relative to CASIM_NO_PROC such that by 0700 UTC, CASIM_NO_PROC has the higher LWP. The peak LWP in SIMPLE_CLOUD occurs around the same time but persists for longer, before decreasing sharply around 1000 UTC. The other difference between the two simulations is in the evolution of the domain mean cloud base

height. SIMPLE_CLOUD maintains an elevated cloud base height compared to CASIM_NO_PROC throughout the simulation period. Between 0530 - 0800 UTC, the mean cloud base height is 60 m higher in SIMPLE_CLOUD, increasing to an average of 140 m higher between 0800 - 1100 UTC.

The differences in the evolution of LWP and cloud base height between the two simulations can be explained through consideration of the effects of droplet sedimentation as a result of gravitational settling. There is no representation of either

sedimentation or warm rain production in SIMPLE_CLOUD, whereas both processes are represented in CASIM_NO_PROC.




However, values of rain water path in CASIM_NO_PROC are typically four orders of magnitude less than the liquid water path, and no precipitation reaches the surface, suggesting that droplet sedimentation and not the warm rain process is responsible for the differences between the simulations. This was confirmed by running a test simulation of CASIM_NO_PROC with droplet sedimentation switched off but autoconversion and accretion left switched on, which effectively yielded the same results as the

5 SIMPLE_CLOUD simulation.

The link between droplet sedimentation and LWP has been explored previously by Bretherton et al. (2007), in the context of nocturnal non-drizzling marine stratocumulus layers in the subtropics. Sedimentation was found to ultimately increase LWP, caused by the removal of liquid water from the entrainment zone near cloud top. In turn this reduces the magnitudes of evaporative cooling and long wave radiative cooling, two processes which control the sinking of relatively dry air from the

10 free troposphere into the cloud layer. Conversely, higher CCN concentrations decrease the mean droplet size and fall speed, reducing sedimentation rates and thus making the cloud more susceptible to the effects of entrainment at the top of the BL. This results in a reduced LWP and a thinner cloud layer for more polluted conditions. We now conduct further analysis of the two MONC experiments to explore whether the results of the present study are consistent with the findings of Bretherton et al. (2007).

Returning to Fig. 12, following completion of the spin-up phase at 0530 UTC, both simulations have a very similar value of LWP. As mentioned earlier, the initial development of the cloud layer during the spin-up phase is strongly dependent on the mechanism of long wave radiative cooling. The lack of droplet sedimentation in SIMPLE_CLOUD does mean however that this experiment is able to maintain a slightly higher liquid water content at cloud top during spin-up relative to CASIM_NO_PROC, with a more sharply defined peak value (see fig 13 compared to fig 7a). Over the following 1.5 hours up to 0700 UTC, the

larger liquid water content in SIMPLE_CLOUD within the entrainment zone promotes stronger evaporative cooling relative to CASIM_NO_PROC, which increases the downward heat flux at cloud top, reduces moisture fluxes and reduces the circulation strength in the BL (fig 14). This results in a slower rate of LWP growth with time relative to CASIM_NO_PROC, such that by 0700UTC, CASIM_NO_PROC has the higher LWP. Thus in CASIM_NO_PROC, the removal of liquid water mass from cloud top due to droplet sedimentation effectively acts to shield the cloud layer to some extent from the effects of entrainment,

allowing LWP to grow faster with time.

It is important to remember that the present study is over land and the simulation period extends into the day time, as opposed to the nocturnal marine BL simulated by Bretherton et al. (2007). Thus it is no surprise that after 0800 UTC, when the fluxes of sensible and latent heat dominate and the surface layer becomes unstable, the effect of sedimentation on LWP starts to break down. The convective vertical mixing associated with the prescribed sensible and latent heat fluxes coincide with the lifting

of the cloud layer and a decrease in LWP, with a more rapid depletion evident in CASIM_NO_PROC. This is consistent with stronger evaporative cooling during mixing associated with the higher LWP around 0730 UTC.

It is interesting to consider what would happen to the evolution of LWP in the absence of surface driven mixing. This is important because, although the mean LLC onset time at Savé is 0300 UTC (around three hours before sunrise), it is notably earlier at other ground-sites (e.g. 0000 UTC in Kumasi, and 2100 UTC at Ile-Ife; Kalthoff et al. 2017). Assuming the

35 sedimentation-entrainment feedback holds true, an earlier LLC onset would allow more time for the effects of sedimentation





to impact LWP before sunrise. To explore this idea, both experiments were re-run with surface fluxes set to zero throughout and with short wave radiation turned off for the duration of the simulation. The forcing of the low-level jet was left unchanged. The results are shown in figure 15. As anticipated, it can clearly be seen that when nocturnal conditions are maintained, CASIM_NO_PROC maintains a higher LWP by around 33% relative to SIMPLE_CLOUD by the end of the simulation period. Based on this analysis, we conclude that the response of the model is consistent with the reasoning of Bretherton et al. (2007).

## 5   Discussion

The numerical experiments performed in this study have shown that droplet sedimentation helps to promote a more heterogeneous cloud layer, with localised regions of both enhanced LWP and reduced LWP within the model domain relative to simulations without droplet sedimentation, whilst also lowering cloud base height. Whilst surface fluxes remain relatively small, in this case prior to 0700 UTC, sedimentation also acts to increase the rate of mean LWP growth within the domain, by buffering the cloud layer from the effects of entrainment-induced evaporative cooling.

Since droplet sedimentation rates are inversely proportional to number concentration, one would expect the effects of sedimentation on both LWP and cloud base height to become more prominent as cloud droplet number concentration (CDNC) reduces. In the case of CASIM_NO_PROC, predicted number concentrations lie in the range 400-700 cm$^{-3}$ at STP, which agrees well with in-situ measurements with median values of around 500 cm$^{-3}$ at STP (J. Taylor, personal communication, 2018). In this section we perform some new experiments to explore the sensitivity to reducing CDNC. We introduce results from a new experiment, CASIM_200, which prescribes the initial CDNC to be 200 cm$^{-3}$. This new simulation produces excessive variability in the LWP field and cloud bases that are too low (figs 16 and 17 respectively), confirming our hypothesis. This was found to be the case even with autoconversion switched off. The depth of the BL in CASIM_200 is also too shallow by the end of the simulation period, by virtue of the increased droplet size and excessive sedimentation velocity. However, mean LWP is slightly lower compared to CASIM_NO_PROC; this is because, around 0830 UTC, cloud base becomes so low it touches the surface and liquid water is removed from the domain. At 200 cm$^{-3}$, this removal of liquid is predominantly due to gravitational settling of cloud droplets as opposed to significant warm rain production. Further reductions in CDNC, down to 100/cc and 50/cc respectively, deplete the LWP even more as a result of an increase in autoconversion. These results, as summarised in table 1, suggest that the effects of droplet size on cloud-top entrainment rates should not be ignored when considering the diurnal cycle of LLCs in the region.

In light of this result, it is pertinent to consider the potential implications of changes in CDNC in terms of cloud radiative effects. Any elevation of CDNC within urban plumes will increase cloud optical depth in a manner that is proportional to CDNC$^{1/3}$ and LWP$^{2/3}$ for shallow clouds. However, the reduced sedimentation associated with the increased CDNC would increase cloud-top entrainment and therefore reduce LWP. Hence any effect of increased optical thickness arising from enhanced aerosol concentrations will to some extent be offset by the sedimentation-entrainment feedback, and is likely to lessen any first order indirect effect.





Our findings also have implications for the diagnosis of aerosol-cloud interactions from satellite data. An adiabatic cloud profile is typically assumed when estimating cloud properties from satellites, but a relevant issue here is the extent to which the adiabatic assumption holds in these low level clouds (Merk et al., 2016). Since satellites view cloud top, it is conceivable that the sedimentation-entrainment effect may well bias retrievals significantly.

5    An important caveat in our results is the prescription of surface fluxes in our simulations; there is no feedback between changes in cloud cover, LWP and the land surface radiation budget. What happens after sunrise in reality is likely to be dependent on such feedbacks, which the present model configuration is not able to capture due to the lack of an interactive land surface scheme. Coupling of MONC to an interactive land surface scheme is needed to be able to comment fully on the impacts of droplet sedimentation and cloud optical depth on the diurnal cycle of these low level clouds.

## 10  6   Conclusions

In this study, large eddy simulations of low level clouds over southern West Africa have been performed with a focus on establishing the sensitivity of the cloud evolution to different treatments of the microphysics. The simulations are constrained and validated using the unprecedented suite of measurements collected during the DACCIWA field campaign in 2016.

Our results reveal that, even for non-precipitating clouds, the evolution of low level clouds over southern West Africa is
sensitive to the effects of droplet sedimentation, suggesting that this mechanism should not be neglected when performing large scale simulations of the region. Sedimentation of droplets acts to remove liquid water from the entrainment zone near cloud top, reducing the magnitude of evaporative cooling during entrainment mixing. This increases the rate of growth of liquid water path during the night time and early morning period. For the conditions of prescribed subsidence and surface fluxes, the simulation best able to reproduce the observations was the one that came closest to matching the observed droplet number
concentrations. Ignoring droplet sedimentation completely reduced variability in liquid water path by around a factor of 2 during the early morning, and also elevated the mean cloud base height by an additional 200 m by the end of the simulation period. Conversely, overestimating sedimentation rates, by virtue of reducing the droplet number concentration by a factor of two or more relative to observed values, caused cloud base to lower to the surface by 0830 UTC, and liquid water path variability to increase around a factor of 2. Both these changes degraded the realism of the model simulation with respect to
the available observations. In all cases, cloud top long wave radiative cooling during the night was found to be crucial for the formation and maintenance of the clouds.

The link between sedimentation and liquid water path has been noted previously in relation to nocturnal non-drizzling marine boundary layer clouds. But the clouds considered in the present study form over land and persist into the day time, which means that the effect of sedimentation can potentially play an important role in regulating the surface radiation budget,
with consequences for the diurnal cycle of the boundary layer in southern West Africa and the circulation of the West African Monsoon. The results of our study suggest the possibility of a complex feedback chain involving aerosols, sedimentation, entrainment, liquid water path and surface energy fluxes. We recommend as part of future work that the experiments performed





as part of this study be repeated using an interactive land surface scheme, to determine the extent to which the sensitivities shown are modified due to feedbacks between cloud cover and the surface heat flux budget.

*Code and data availability.* The observational data used in this paper can be accessed upon request at http://baobab.sedoo.fr/DACCIWA. The MONC, CASIM and SOCRATES codes are maintained by the Met Office and accessible via the Met Office Science Repository Service

(https://code.metoffice.gov.uk/):

MONC branch: main/branches/dev/chrisdearden/r4366_dacciwa_socrates_vn0.8_vn0.9_part2

CASIM branch: casim/branches/dev/chrisdearden/r4323_casim_vn10.8_monc_fixes

For further details, please contact Christopher Dearden (c.dearden@leeds.ac.uk) or Adrian Hill (adrian.hill@metoffice.gov.uk).

*Competing interests.* The authors declare that they have no conflict of interest.

*Acknowledgements.* The research leading to these results has received funding from the European Union 7th Framework Programme (FP7/2007-2013) under Grant Agreement no. 603502 (EU project DACCIWA: Dynamics-aerosol-chemistry-cloud interactions in West Africa).The authors would like to acknowledge Norbert Kalthoff, Bianca Adler, Karmen Babic, Fabienne Lohou, Cheikh Dione, Marie Lothon and Xabier Pedruzo Bagazgoitia for their role in producing the observations presented in this paper, and for helpful discussions at the DACCIWA project meeting in Karlsruhe, Germany, 24-27 October 2017. This work used the ARCHER UK National Supercomputing

Service (http://www.archer.ac.uk) and the JASMIN service (http://www.jasmin.ac.uk).



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





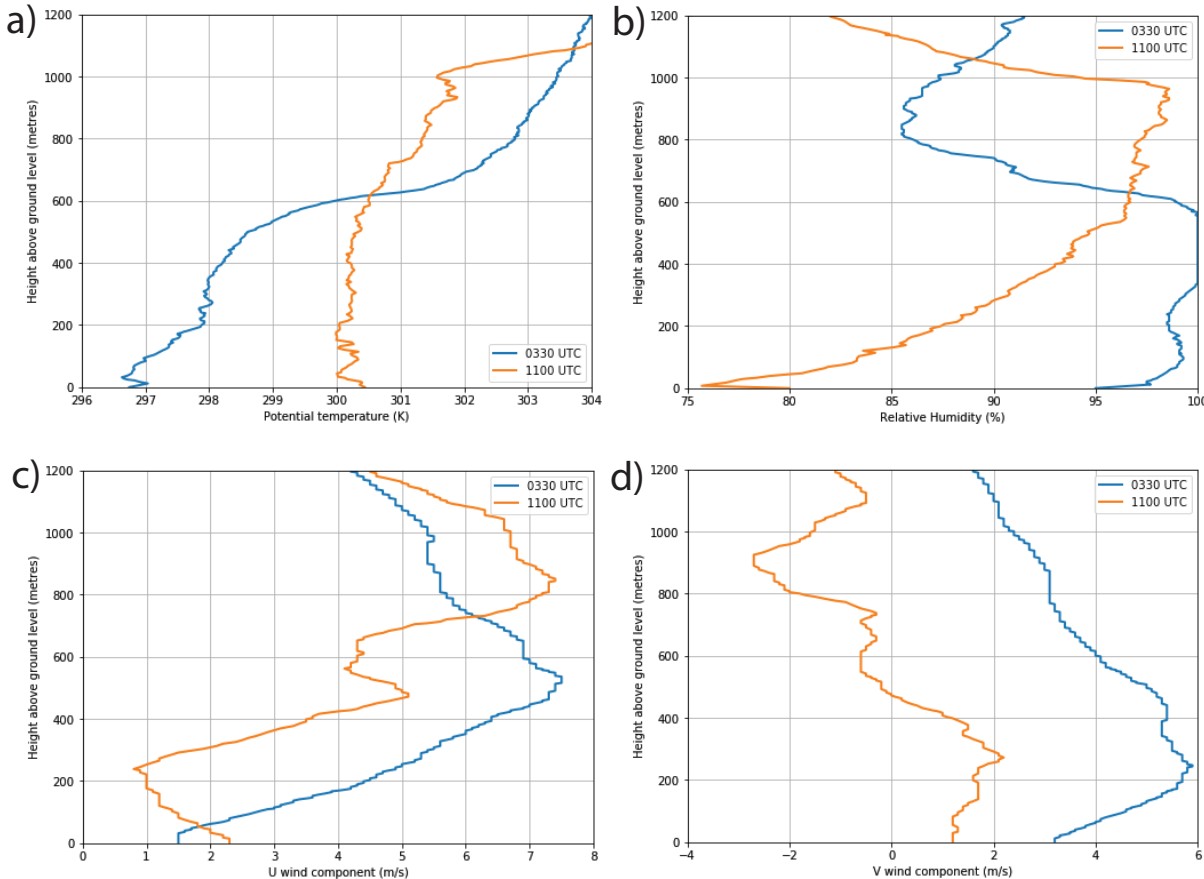

**Figure 1.** Profiles of a) potential temperature, b) relative humidity, c) u wind component and d) v wind component from radiosondes launched at Savé on 5 July 2016 at 0330 UTC (blue) and 1100 UTC (orange).



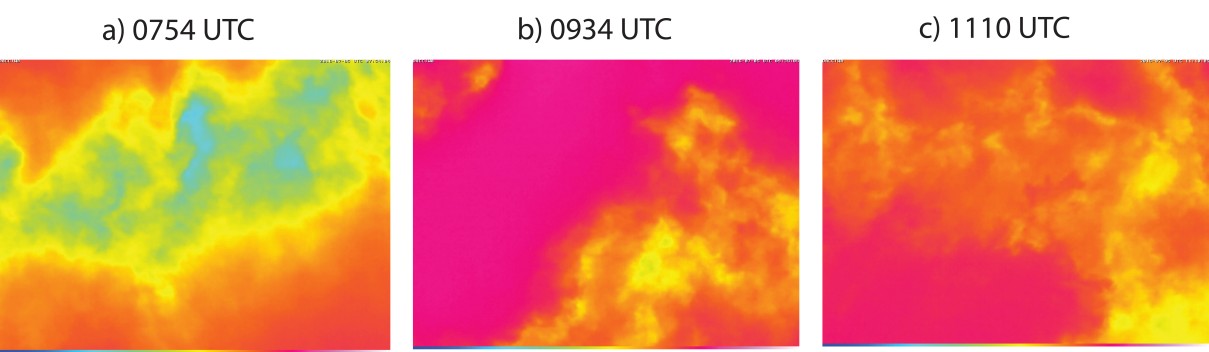

a) 0754 UTC          b) 0934 UTC          c) 1110 UTC

**Figure 2.** Images from the Infrared cloud camera at Savé on 5 July 2016 (Handwerker et al., 2016) for the times indicated.



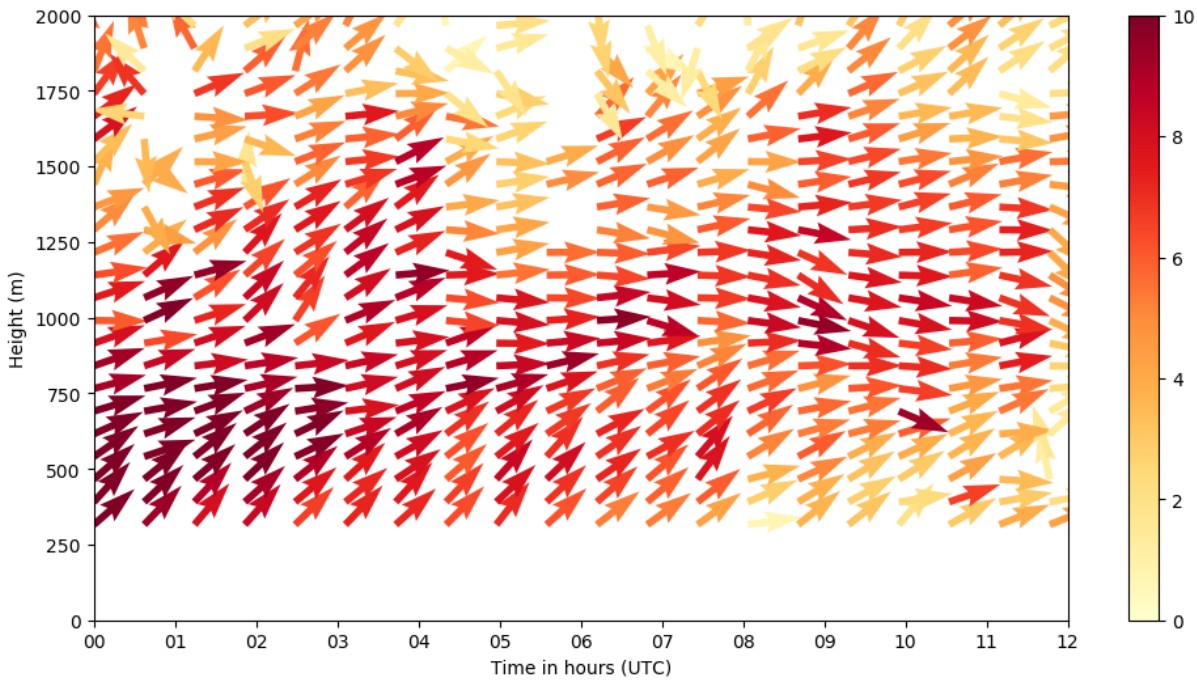

**Figure 3.** Time-height plot showing the vertical profile of the horizontal wind at Savé on 5 July 2016, from the Ultra-High Frequency wind profiler (Derrien et al., 2016). Wind vectors are normalised and indicate the direction of the horizontal flow; shading indicates the wind speed (m s$^{-1}$).




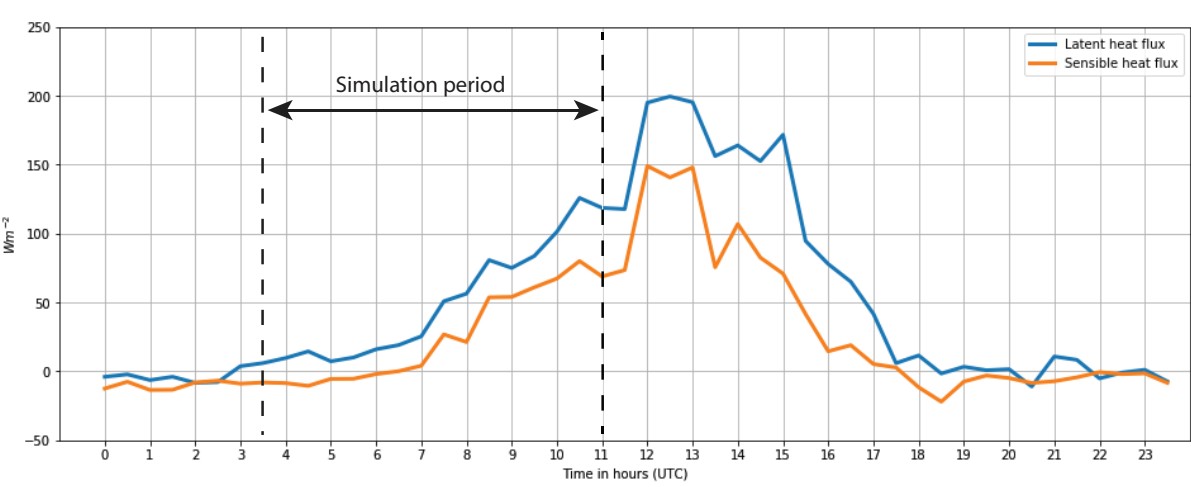

**Figure 4.** Timeseries of latent heat flux (blue) and sensible heat flux (orange) from the Savé ground-site on 5 July 2016 (Kohler et al., 2016).





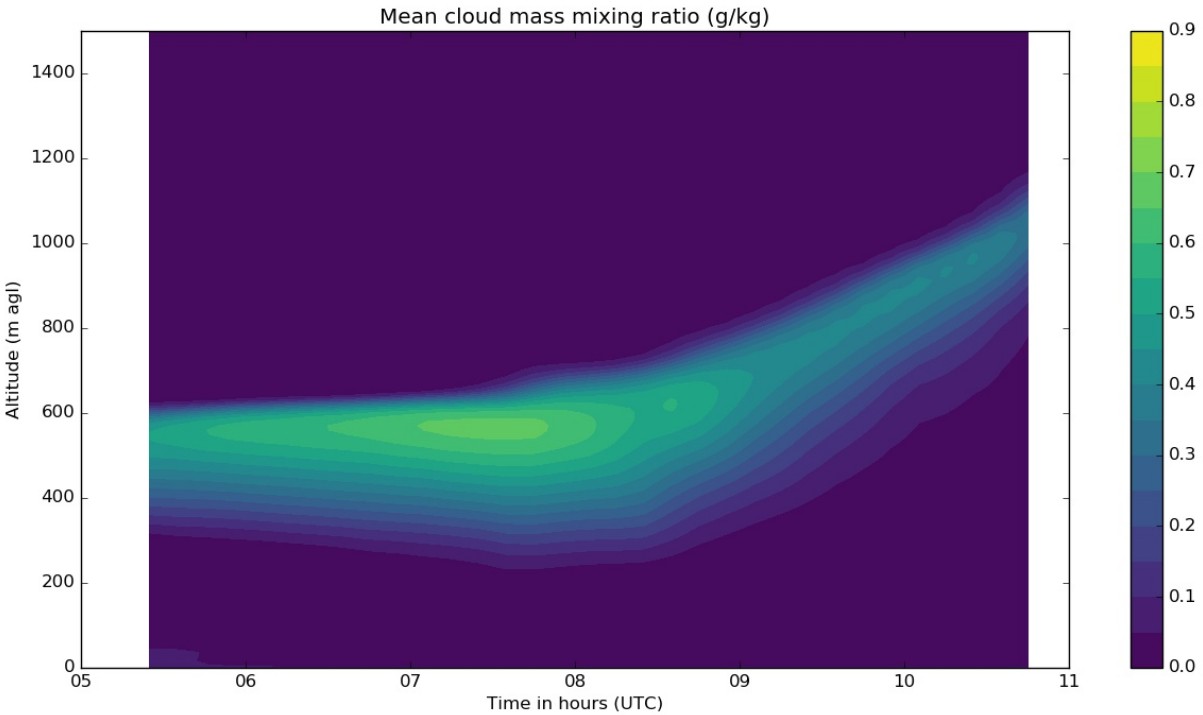

**Figure 5.** Time-height plot of the mean cloud mass mixing ratio (g kg$^{-1}$) within the model domain from the CASIM_NO_PROC experiment. Values are calculated as temporal means every 10 minutes.





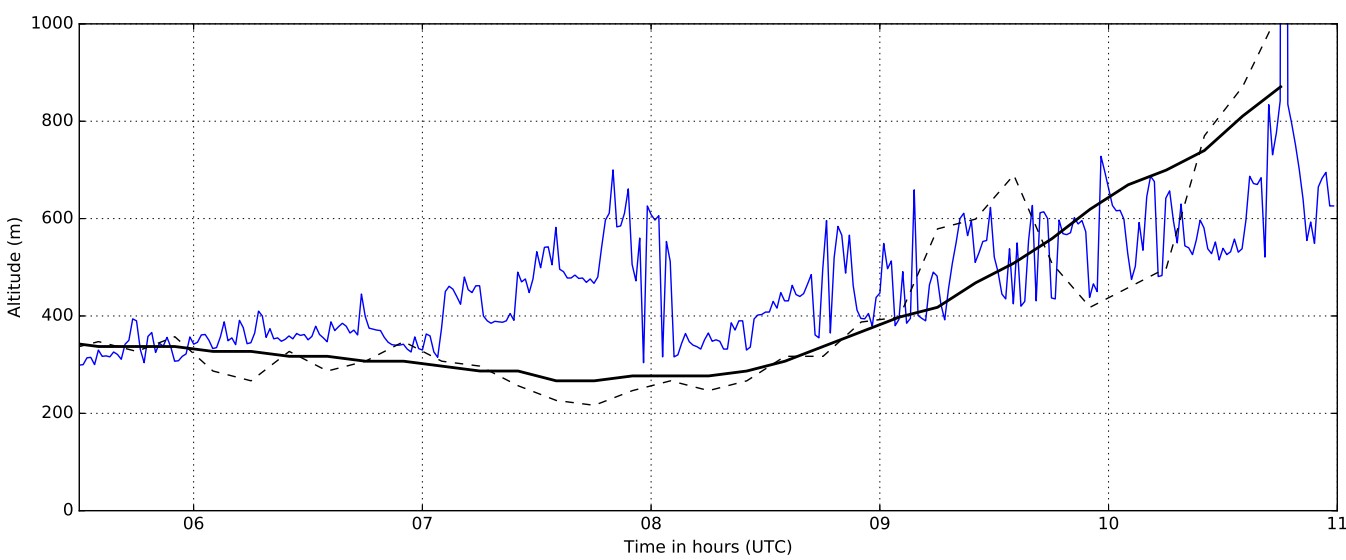

**Figure 6.** Timeseries of cloud base height at Savé on 5 July 2016 (blue) derived from ceilometer measurements (Handwerker et al., 2016), and 10 minute averages of cloud base height diagnosed from CASIM_NO_PROC using a threshold cloud liquid water mass mixing ratio of 0.1 g kg$^{-1}$. Solid black line - domain mean value; dashed black line - the value at the centre of the model domain.





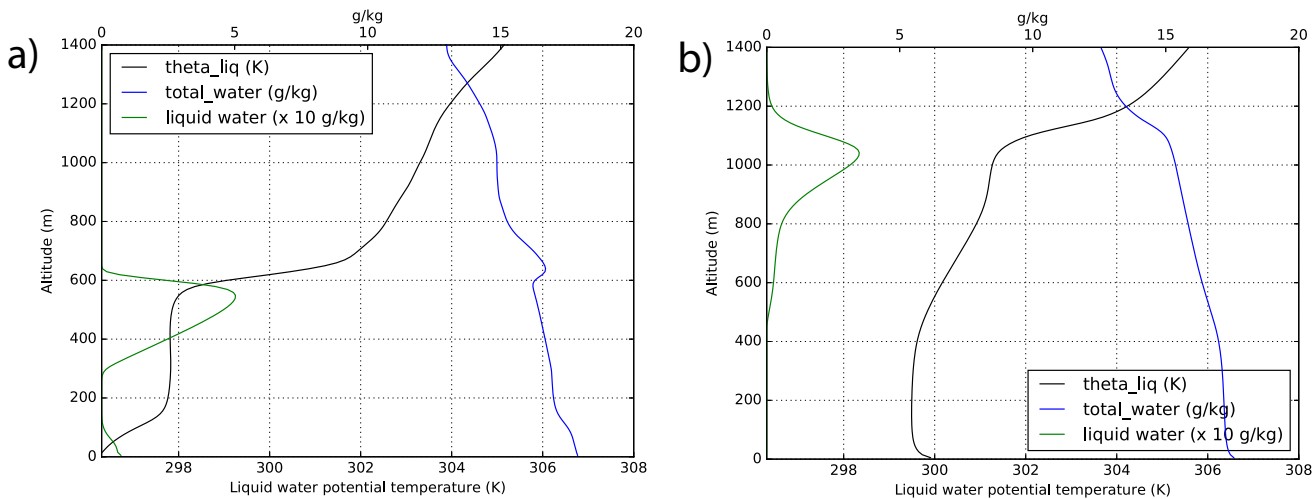

**Figure 7.** Vertical profiles of liquid water potential temperature (K; black lines), total water mass mixing ratio (g kg$^{-1}$; blue lines) and liquid water mass mixing ratio (x 10 g kg$^{-1}$; green lines) diagnosed at a) 0530 UTC and b) 1100 UTC from the CASIM_NO_PROC experiment.





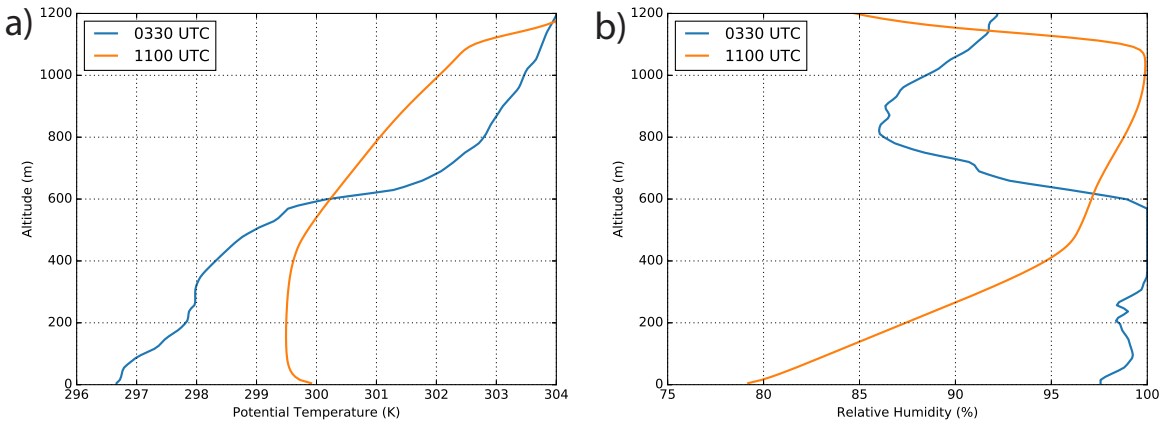

**Figure 8.** Simulated domain-average vertical profiles of a) potential temperature and b) relative humidity from the CASIM_NO_PROC simulation, calculated at 0330 UTC (blue) and 1100 UTC (orange).

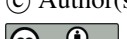



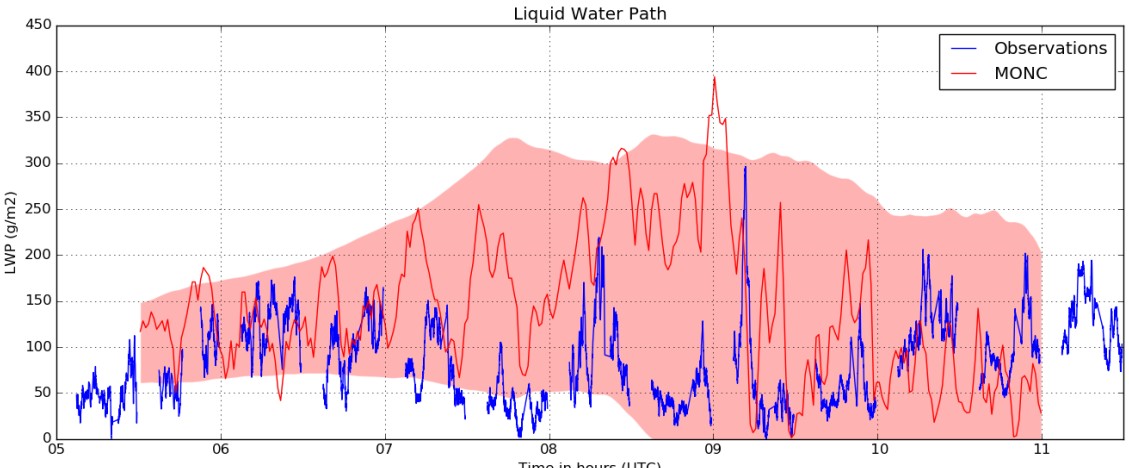

**Figure 9.** Comparison of LWP timeseries at Savé from 5 July 2016 (blue) as measured by the microwave radiometer (Wieser et al., 2016), with simulated LWP from CASIM_NO_PROC, showing the evolution of LWP at the centre of the model domain (red line) and the LWP variability within the whole domain (red shading), expressed as +/- 2 standard deviations from the domain mean value.





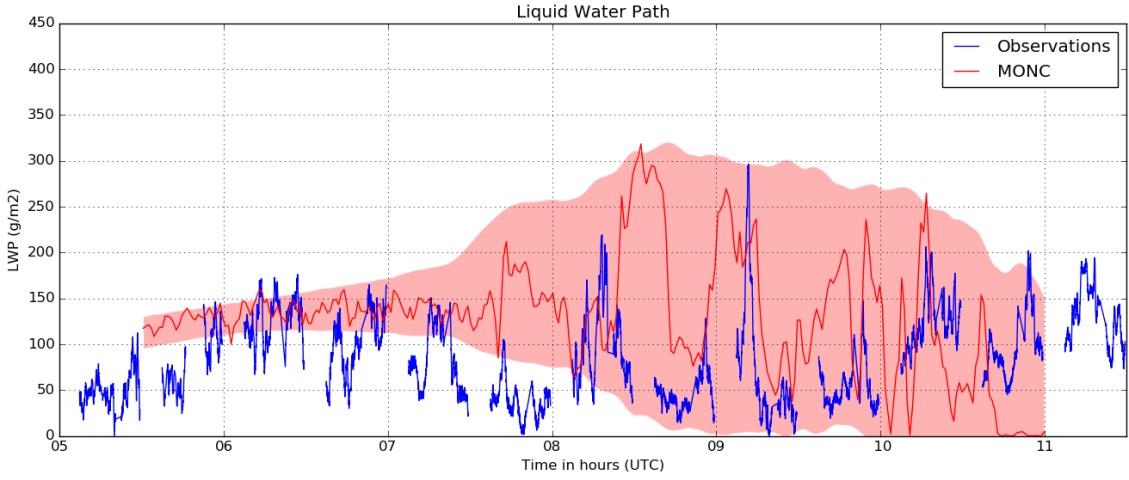

**Figure 10.** As Fig 9 but for the SIMPLE_CLOUD experiment.





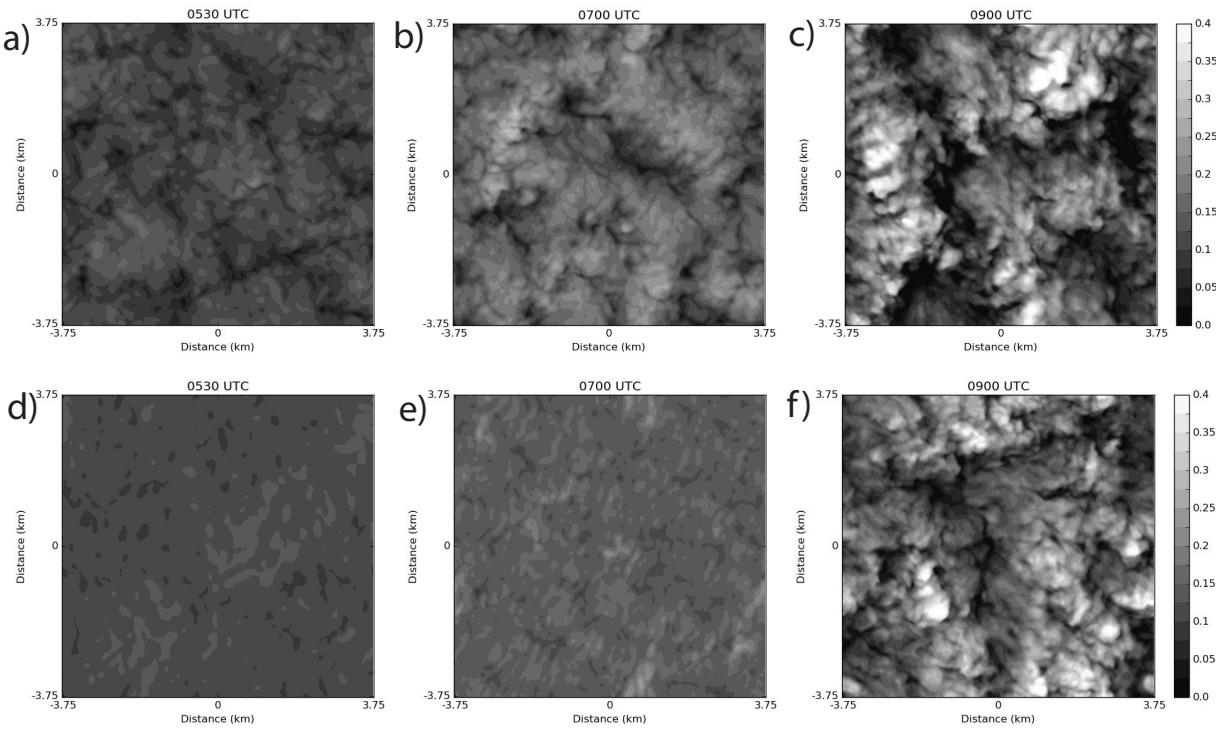

**Figure 11.** Maps showing the spatial distribution of LWP (kg m$^{-2}$) within the model domain at 0530, 0700 and 0900 UTC for CASIM_NO_PROC (a-c; top row) and SIMPLE_CLOUD (d-f; bottom row).





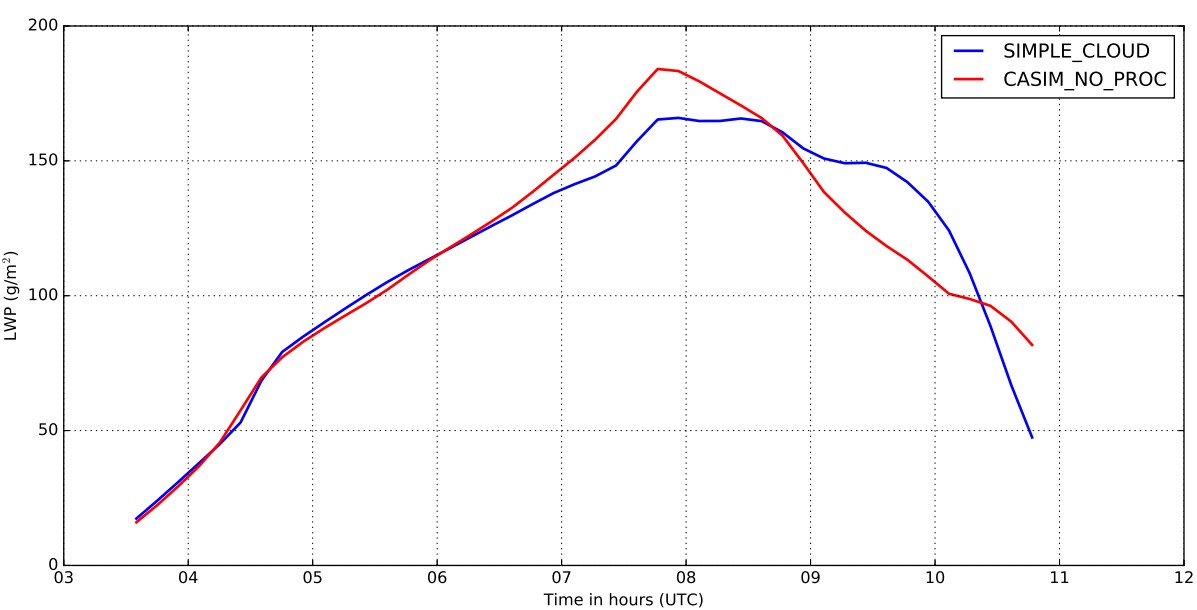

**Figure 12.** Timeseries of simulated LWP (g m$^{-2}$) from CASIM_NO_PROC (red) and SIMPLE_CLOUD (blue). LWP is calculated from 200 m to the top of the model domain, in order to ignore the thin fog layer near the surface that develops during the spin-up period and dissipates around 0630 UTC.





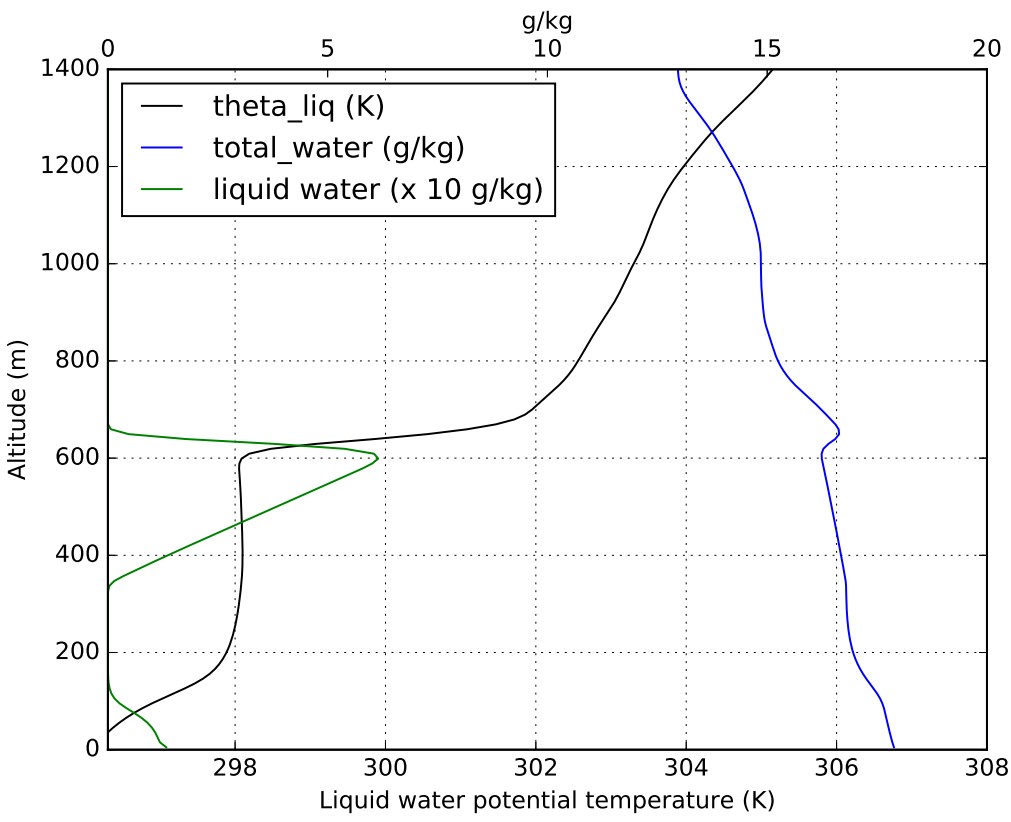

**Figure 13.** As Fig 7 but for SIMPLE_CLOUD at 0530 UTC.





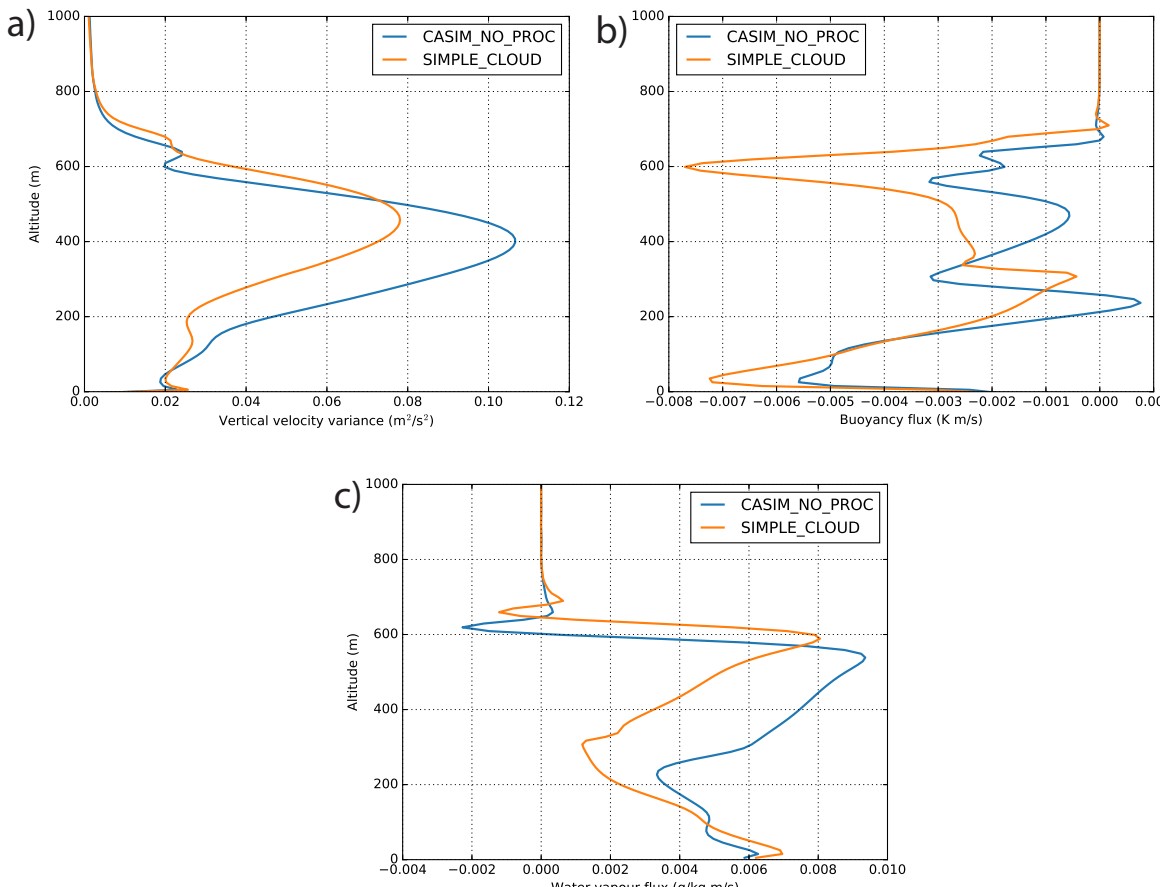

**Figure 14.** Domain average vertical profiles of a) vertical velocity variance ($m^2 s^{-2}$) b) buoyancy flux ($K\ m\ s^{-1}$) and water vapour flux ($g\ kg^{-1}\ m\ s^{-1}$), calculated as temporal means between 0530 UTC - 0700 UTC for CASIM_NO_PROC (blue) and SIMPLE_CLOUD (orange).





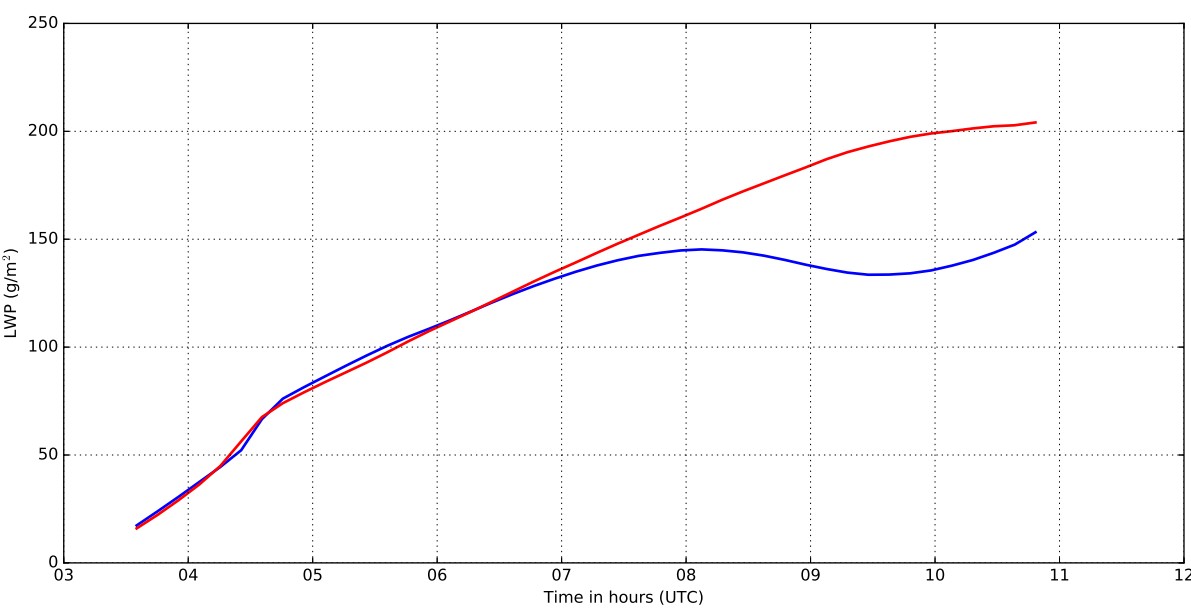

**Figure 15.** As Fig 12 but with short wave radiation disabled and surface fluxes set to zero throughout the simulation period.





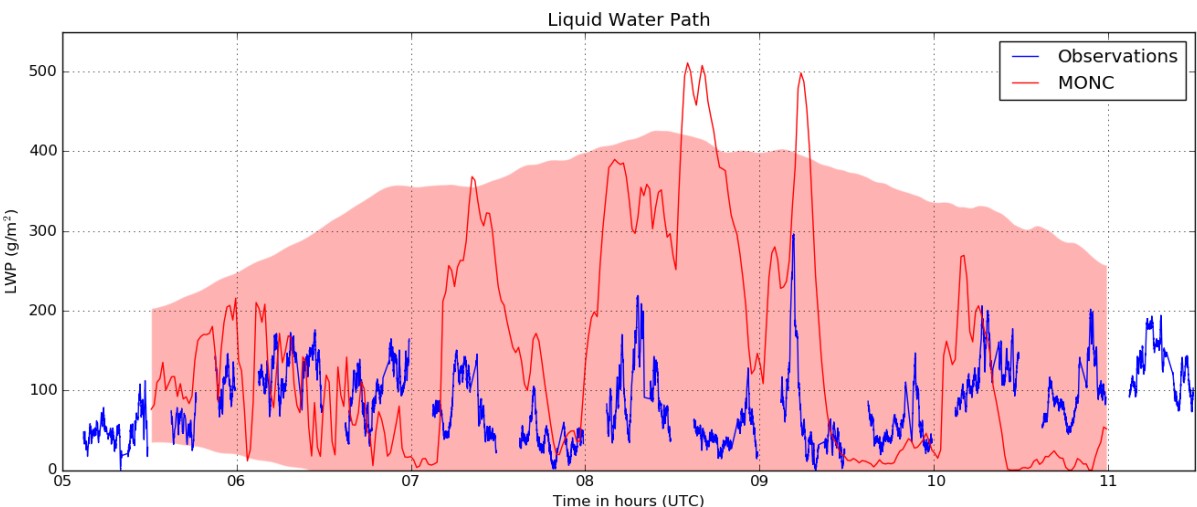

**Figure 16.** As Fig 9 but for CASIM_200.



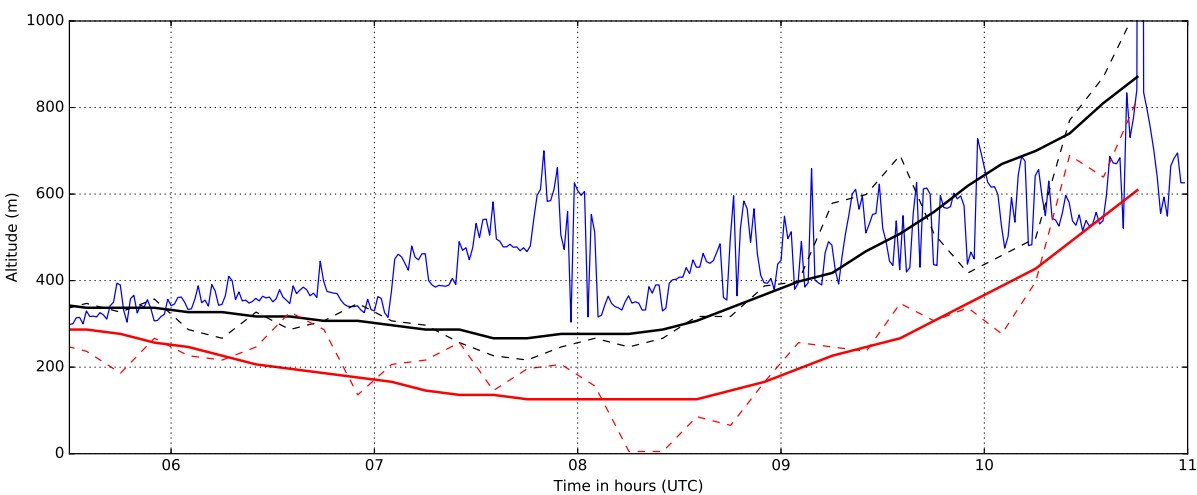

**Figure 17.** As Fig 6 with results from CASIM_200 added (in red).



**Table 1.** Table showing the mean values of liquid water path (g m$^{-2}$), rain water path (g m$^{-2}$) and surface precipitation rate (mm h$^{-1}$) calculated between 0600 - 0800 UTC and 0800 - 1000 UTC for five different simulations, listed in order of increasing rates of droplet sedimentation achieved by reducing droplet number.

| | 0600 - 0800 UTC | | | 0800 - 1000 UTC | | |
|---|---|---|---|---|---|---|
| | LWP | RWP | precip rate | LWP | RWP | precip rate |
| SIMPLE_CLOUD | 140.75 | 0 | 0 | 156.66 | 0 | 0 |
| CASIM_NO_PROC | 150.08 | 0.014 | 0 | 150.65 | 0.015 | 0 |
| CASIM_200 | 133.96 | 0.16 | 0.0015 | 129.50 | 0.17 | 0.0017 |
| CASIM_100 | 129. 49 | 0.24 | 0.004 | 122.47 | 0.33 | 0.0069 |
| CASIM_50 | 90.58 | 0.44 | 0.025 | 83.29 | 0.59 | 0.015 |