# Peer review of "The role of droplet sedimentation in the evolution of low level clouds over southern West Africa"

_Atmospheric Chemistry and Physics, 2018_

## Referee Comment (RC1) · Anonymous Referee #1 · 11 May 2018

Review of "The role of droplet sedimentation in the evolution of low level clouds over southern West Africa", submitted to ACP by Dearden et al. Paper number ACP-2018-269

This paper describes simulations of low clouds that form and deepen over the diurnal cycle during the monsoon season in West Africa. Enabling/disabling sedimentation of cloud droplets has a substantial impact on the evolution of the cloudy boundary layer.

Assessment:

This well-written paper does a good job of describing the sensitivity of low cloud to the changes through the diurnal cycle and to the various sensitivity studies with the

model. Isolating cloud droplet sedimentation as key to the differences is a nice accomplishment. However, I have some concerns (outlined in more detail below) that the presence/absence of cloud droplet sedimentation during the spin-up of turbulence and convection at the start of the simulation may have a larger-than-expected impact on the simulations. As a result, I would ask the authors to branch a pair (or a full set) of simulations with/without sedimentation from a spun-up initial condition. I expect that the impact of sedimentation will be more modest in this case but still non-negligible and that the revised paper will be suitable for publication in ACP.

I also make some suggestions regarding the consolidation of figures, as I find seventeen figures to be a lot for a paper of this length, and I sometimes think that it's helpful to look at the relationship of related fields or simulations in multi-panel figures. As this is a stylistic concern, the authors may ignore these suggestions if they wish.

Recommendation: Major revisions.

Major comment:

1. A close reading of Bretherton et al (2007, section 3, paragraph 7) reveals that each of their simulations proceeded identically without sedimentation for two hours before sedimentation was enabled in two of the three simulations. My impression is that the initial absence of turbulence and convection in the boundary layer allows the presence/absence of sedimentation to have an outsize effect when there is no compensating water flux into the inversion. With a cloud droplet sedimentation speed of 5 mm/s (which seems about right for $q_c=0.5$g/kg and $N_d=500$/cm3), the cloud layer would subside by one grid level (10m) over 2000 seconds, or about a half hour. If it takes the turbulence about a half hour to spin up at the start of the simulation, this sedimentation might account for a significant part of the difference in inversion height between CASIM_NO_PROC and SIMPLE_CLOUD at 0530, which I estimate as 20m. Isolating the effect of sedimentation by turning on/off sedimentation in an already turbulent boundary layer would eliminate this uncertainty and strengthen the study in my

view.

For the present paper, I would suggest using the CASIM_NO_PROC simulation as the control for the first two hours since it does a good job of maintaining the observed LWP and then branching all the simulations from that point. If it's easier, use CASIM_NO_SED in place of SIMPLE_CLOUD since the restart file may not want the microphysics scheme to be switched. If the comparisons of early morning cloud and boundary layer properties should stay at 0530 hours, perhaps the start of the spin-up simulation should be pushed back to 0130 hours.

Perhaps, this won't have a huge effect, but it's worth checking.

1a. The results of Ackerman et al (2004) suggest that the effect of sedimentation may be more notable in boundary layers with dry air above the PBL. Toll et al (2017, GRL, https://doi.org/10.1002/2017GL075280) note an LWP decrease in ship and volcano tracks in non-precipitating clouds with dry air aloft (bottom of their figure 2) that could be related to the droplet sedimentation effect. It would be interesting to see that same effect here in a boundary layer with a very weak moisture jump across the inversion. The effect on entrainment could be compared to the parameterization in section 6 of Bretherton et al.

2. Comparing cloud base observations with simulations: Instead of plotting the time series of a single model column for comparison with the observed cloud base height, I would suggest plotting three quantities than span the range of cloud base heights in the model:

+ the inversion height (roughly the top of the stratocumulus cloud),

+ the median cloud base height of cloudy column (roughly stratocumulus cloud base),

+ the lowest cloud base (where cloud fraction first reaches 1% or so) or the LCL of the subcloud layer. This roughly gives the cumulus cloud base in a decoupled boundary layer.

The observed cloud base height will nicely follow these lines, I think, with the lowest model cloud base capturing the low observed cloud base heights after 1000 hours. The presence of fog makes the computation of the lowest cloud base/LCL a bit more complicated, but I would suggest lowest non-fog cloud base. Note that the divergence of the median cloud base height from the lowest cloud base/LCL is a good indicator of decoupling.

Minor comments (5/26 means page 5, line 26):

5/1: What time is sunrise? I'm not sure it's crucial, but I felt myself wondering as I read the manuscript.

8/20: Suggested re-wording: "... by virtue of the effect of increased droplet size and excessive sedimentation velocity on entrainment."

9/30: Suggested re-wording: "... and _possibly_ the circulation of the West African monsoon."

Figures: These are stylistic suggestions, but I feel that grouping these many figures into fewer multi-panel figures could help the reader interrogate their meaning more easily. Feel free to ignore this advice if you wish.

+ Fig 0: An additional figure with a map-like image would be helpful for the reader who hasn't thought so carefully about clouds over Africa. How about a visible geostationary satellite image from 11Z showing the breakup of the cloud along with the locations of the coast, Sav\'e, Lome and the transect?

+ Fig 2: A bigger colorscale would be helpful

+ Figs 3-4: Could figures 3 and 4 be stacked? It would be cool to see Fig 3 extended over the full 24 hours and see the re-formaton of the jet in the evening. This would also let the reader clearly see the result of the strong afternoon surface buoyancy flux on the wind field.
+ Figs 5-6: Could these be stacked?

- Fig 5: I think it would be helpful to NaN (make blank) the regions where qc==0.

- Fig 6: Note major comment 2 above. If the cloud-free regions are white in figure 5, these lines could even be superimposed on figure 5, though that might be too much.

+ Figs 7,13: Could these be stacked with an additional panel for the 1100 UTC version of SIMPLE_CLOUD? Could the lowest cloud base, median cloud base and inversion height be marked as dashed lines.

+ Fig 8: If the authors think it's helpful, could the observations from figure 1 be added as dashed lines?

+ Figs 9, 10, 16: Could these be stacked as a three-panel figure? I felt the need to flip back and forth to compare the different versions of this figure.

+ Figs 12, 15: Could the lines in figure 15 be added as dashed lines in figure 12 if that's not too distracting?

Table 1: Could the cloud droplet number concentration be added to the table? For the run with predicted droplet concentration, a range of values could be given that could be different between the two times if appropriate.

---

## Referee Comment (RC2) · Anonymous Referee #2 · 28 May 2018

Using an observationally well-characterized case in southern West Africa, the role of sedimentation of cloud droplets in determining liquid water path and heights of low-level clouds, once established, is illustrated using large-eddy simulation and microphysical parameterizations with and without sedimentation. Controls by cloud drop number concentration (and drop size) on the extent to which sedimentation is effective in determining cloud height and water path are also discussed.

This is an important paper, extending earlier work on marine clouds and potential cloud-aerosol interactions related to sedimentation to land clouds. Although many questions remain, especially related to the roles of interactive surface fluxes of heat

and moisture, which are not considered here, the paper advances knowledge of low-clouds in a region where they play an important role in the regional surface radiation balance and may be subject to strong aerosol interactions.

The paper is generally well written. While I agree with RC 1 about consolidating figures, the study offers the opportunity to illustrate some of the physical mechanisms at play in more detail, and I suggest the authors consider doing so. Specifically,

1. On Fig. 6, characterize the cloud base altitude for SIMPLE_CLOUD as has been done for CASIM_NO_PROC.

2. A figure illustrating the different mixing ratio profiles for the cases in Table 1 would help to visualize the corresponding differences in sedimentation in these cases. A figure showing some measure of droplet size would also be helpful.

3. What are the units of the field shown in Fig. 2?

4. The importance of long-wave radiative cooling is discussed for three features of the simulation: (1) cloud formation and maintenance (p. 5, ll. 25-28; p. 7, ll. 15-18; p. 9, l. 25); (2) formation of stable layer near surface overnight (pp. 5-6); and (3) reduced long-wave cooling near cloud top due to sedimentation (p. 7, l. 9). Figures illustrating radiative cooling rates would illustrate these points effectively. Also, with sedimentation, both radiative and evaporative cooling are reduced near cloud top. A figure comparing these rates would be very useful in understanding the relative roles of the two processes.

---

## Author Comment (AC1) · 26 Jul 2018

The authors would like to thank both reviewers for their helpful comments. Below we address each comment in turn, starting with reviewer #1. Our responses are highlighted in bold.

Response to comments by Reviewer #1

Major comments:

*1. A close reading of Bretherton et al (2007, section 3, paragraph 7) reveals that each of their simulations proceeded identically without sedimentation for two hours before sedimentation was enabled in two of the three simulations. My impression is that the initial absence of turbulence and convection in the boundary layer allows the presence/absence of sedimentation to have an outsize effect when there is no compensating water flux into the inversion. With a cloud droplet sedimentation speed of 5 mm/s (which seems about right for qc=0.5g/kg and Nd=500/cm3), the cloud layer would subside by one grid level (10m) over 2000 seconds, or about a half hour. If it takes the turbulence about a half hour to spin up at the start of the simulation, this sedimentation might account for a significant part of the difference in inversion height between CASIM_NO_PROC and SIMPLE_CLOUD at 0530, which I estimate as 20m. Isolating the effect of sedimentation by turning on/off sedimentation in an already turbulent boundary layer would eliminate this uncertainty and strengthen the study in my view.*
*        For the present paper, I would suggest using the CASIM_NO_PROC simulation as the control for the first two hours since it does a good job of maintaining the observed LWP and then branching all the simulations from that point. If it's easier, use CASIM_NO_SED in place of SIMPLE_CLOUD since the restart file may not want the microphysics scheme to be switched. If the comparisons of early morning cloud and boundary layer properties should stay at 0530 hours, perhaps the start of the spin-up simulation should be pushed back to 0130 hours.*
*        Perhaps, this won't have a huge effect, but it's worth checking.*

**We have followed the reviewer's advice by re-running the test simulation with sedimentation disabled within an already turbulent boundary layer. This was achieved by first re-running the CASIM_NO_PROC simulation to generate a checkpoint restart file after 90 minutes of simulation time (the time taken for the model to spin-up from the initial conditions, based on a timeseries plot of the maximum vertical velocity within the domain; see Figure below). The model was then restarted from this 90 minute checkpoint file, but with sedimentation turned off whilst all other settings were kept the same, to produce the new CASIM_NO_SED experiment.**

[Figure]

**To assess the impact on our results, we have reproduced some key plots from the original manuscript, replacing SIMPLE_CLOUD with CASIM_NO_SED. The findings are summarised as follows.**

**Regarding the comparison with liquid water path (LWP) observations, the new CASIM_NO_SED simulation still underestimates the variability in LWP between 0530 and 0700 hours compared to the observations at Save (see figure below), as was the case with SIMPLE_CLOUD (cf figure 10 in the original manuscript).**

[Figure]

*Comparison of LWP timeseries at Savé from 5 July 2016 (blue) as measured by the microwave radiometer (Wieser et al., 2016), with simulated LWP from CASIM_NO_SED, showing the evolution of LWP at the centre of the model domain (red line) and the LWP variability within the whole domain (red shading), expressed as +/- 2 standard deviations from the domain mean value.*

**The plot below compares the timeseries of the mean LWP in the domain for both CASIM_NO_PROC and CASIM_NO_SED. The trend in CASIM_NO_SED is much the same as it was in SIMPLE_CLOUD (cf Figure 12 in the original manuscript).**

[Figure]

**Timeseries of simulated LWP (g m −2 ) from CASIM_NO_PROC (red) and CASIM_NO_SED (blue)**

Finally, the spatial distribution of LWP in CASIM_NO_SED still exhibits the same overly homogeneous structure as SIMPLE_CLOUD relative to CASIM_NO_PROC at 0530 UTC and 0700 UTC, as shown in the plot below (cf Figure 11 in the original manuscript).

[Figure]

*Maps showing the spatial distribution of LWP (kg m −2 ) within the model domain at 0530, 0700 and 0900 UTC for CASIM_NO_PROC (a-c; top row) and CASIM_NO_SED (d-f; bottom row).*

Based on this assessment, we conclude that the suppression of sedimentation during the spin-up of turbulence in the original SIMPLE_CLOUD simulation has negligible impact on our conclusions. Nevertheless, for consistency with Bretherton et al 2007, we have decided to update the manuscript so that we now include results from the new CASIM_NO_SED simulation in the revised manuscript instead of the original SIMPLE_CLOUD test experiment. No changes were made to the CASIM_200, CASIM_100 and CASIM_50 simulations for the revised manuscript.

Note that whilst setting up the CASIM_NO_SED experiment, we also investigated the use of an earlier model start time as suggested by the reviewer. Initialising both CASIM_NO_PROC and CASIM_NO_SED from 0230 UTC rather than 0330 UTC did not have any appreciable impact on the relative difference in LWP between the control and test simulation, but it did result in an overestimation of the liquid water path compared to the observations at around 0500 hours in the control simulation. For this reason, we decided to stick with the original initialisation time of 0330 UTC.

We would also like to point out that by replacing SIMPLE_CLOUD with CASIM_NO_SED, it was necessary to make a number of relatively small changes to the manuscript for consistency, mainly in the Introduction and Model Description sections. With this in mind, we have produced a version of the revised manuscript with track changes, where all differences with respect to the original manuscript are clearly highlighted.

*1a. The results of Ackerman et al (2004) suggest that the effect of sedimentation may be more notable in boundary layers with dry air above the PBL. Toll et al (2017, GRL, https://doi.org/10.1002/2017GL075280) note an LWP decrease in ship and volcano tracks in non-precipitating clouds with dry air aloft (bottom of their figure 2) that could be related to the droplet sedimentation effect. It would be interesting to see that same effect here in a boundary layer with a very weak moisture jump across the inversion. The effect on entrainment could be compared to the parameterization in section 6 of Bretherton et al.*

**The authors agree that it would certainly be interesting to explore the effects of sedimentation in a boundary layer with a weaker moisture jump across the inversion. However, the focus of the present study is on a single DACCIWA case study where the humidity and temperature profiles are constrained from observations. We feel that to perform extra sensitivity simulations of this nature would extend the present study beyond its original scope, and on this basis we feel this would be more appropriate as part of a separate follow-on study.**

*2. Comparing cloud base observations with simulations: Instead of plotting the time series of a single model column for comparison with the observed cloud base height, I would suggest plotting three quantities than span the range of cloud base heights in the model:*
*+ the inversion height (roughly the top of the stratocumulus cloud),*
*+ the median cloud base height of cloudy column (roughly stratocumulus cloud base),*
*+ the lowest cloud base (where cloud fraction first reaches 1% or so) or the LCL of the subcloud layer. This roughly gives the cumulus cloud base in a decoupled boundary layer.*

*The observed cloud base height will nicely follow these lines, I think, with the lowest model cloud base capturing the low observed cloud base heights after 1000 hours. The presence of fog makes the computation of the lowest cloud base/LCL a bit more complicated, but I would suggest lowest non-fog cloud base. Note that the divergence of the median cloud base height from the lowest cloud base/LCL is a good indicator of decoupling.*

**The reviewer implies that we are only plotting the time series of a single model column for comparison against the observed cloud base height, but that is not strictly true. We plot two timeseries from the model – the first is the domain mean cloud base height, and the second is the cloud base height from the column at the centre of the model domain. Together, we feel these two quantities provide a satisfactory representation of the simulated cloud base height and its variation with the model. The main purpose of the Figure is to characterize the representation of cloud base height in the model with respect to the observations, as opposed to diagnosing the coupling state of the boundary layer. As such we feel that including the simulated inversion height in the same plot as well could potentially be confusing to the reader. On balance we have decided to stick with our original approach, but we have also overlaid timeseries plots from CASIM_NO_SED as well, as suggested by reviewer #2.**

Minor comments (5/26 means page 5, line 26):

*5/1: What time is sunrise? I'm not sure it's crucial, but I felt myself wondering as I read the manuscript.*
**Sunrise was at 0537 UTC. This is stated in the original manuscript on page 5, line 22.**

*8/20: Suggested re-wording: "... by virtue of the effect of increased droplet size and excessive sedimentation velocity on entrainment."*
**Done.**

*9/30: Suggested re-wording: "... and _possibly_ the circulation of the West African monsoon."*
**Done.**

*Figures: These are stylistic suggestions, but I feel that grouping these many figures into fewer multi-panel figures could help the reader interrogate their meaning more easily. Feel free to ignore this advice if you wish.*

**Thank you for the ideas to help consolidate the figures. We have strived to implement these where possible; see individual responses below.**

*+ Fig 0: An additional figure with a map-like image would be helpful for the reader who hasn't thought so carefully about clouds over Africa. How about a visible geostationary satellite image from 11Z showing the breakup of the cloud along with the locations of the coast, Save, Lome and the transect?*

**We have included a new Figure (Figure 1 in the revised manuscript, and reproduced below) showing the 0.6 micron visible channel from the Meteosat-10 geostationary satellite. It reveals the cloud structure over southern West Africa at 1012 UTC on 5th July 2016, with borders and coastlines highlighted along with the locations of Save and Lome, labelled 'A' and 'B' respectively in the figure. Note that the satellite image represents all cloud, not just the low-level cloud, but nevertheless the extent of the cloud cover over the Guinea coast is still evident.**

[Figure]

*+ Fig 2: A bigger colorscale would be helpful.*

**It is difficult to enlarge the color scale since it is embedded within the images obtained from the infrared camera and cannot be plotted separately. Instead we have added the following text to the Figure caption: "The images from the camera are coded in RGB colors (red, green, and blue), providing a qualitative estimate of cloud cover during the day and night. The image colour is dependent on the emissivity of the sky and consequently on the brightness temperature, such that red indicates warm and blue cold." We have also made a correction to the reference in the caption – rather than Handwerker et al 2016, it now reads 'Derrien et al 2016'.**

*+ Figs 3-4: Could figures 3 and 4 be stacked? It would be cool to see Fig 3 extended over the full 24 hours and see the re-formaton of the jet in the evening. This would also let the reader clearly see the result of the strong afternoon surface buoyancy flux on the wind field.*

**Thanks for this idea; we have stacked the two figures in the revised manuscript as suggested, using an extended x-axis for both plots to show the re-formation of the jet in the evening. The result is shown below:**

[Figure]

*+ Figs 5-6: Could these be stacked?*
*- Fig 5: I think it would be helpful to NaN (make blank) the regions where qc==0.*

**Both suggestions above have been incorporated into the revised manuscript, to form the new two-panel Figure 5.**

*- Fig 6: Note major comment 2 above. If the cloud-free regions are white in figure 5, these lines could even be superimposed on figure 5, though that might be too much.*
**We chose to keep the cloud base height plots as a separate sub-figure in the end (Figure 5b in the revised manuscript), since we couldn't find an acceptable way to make the lines stand out against the shaded contours in Figure 5a.**

*+ Figs 7,13: Could these be stacked with an additional panel for the 1100 UTC version of SIMPLE_CLOUD? Could the lowest cloud base, median cloud base and inversion height be marked as dashed lines.*
**We have stacked Figs 7 and 13, and included the additional panel for the 1100 UTC version of CASIM_NO_SED, as suggested. See Figure 6 in the revised manuscript.**

*+ Fig 8: If the authors think it's helpful, could the observations from figure 1 be added as dashed lines?*
**Done; see Figure 7 in the revised manuscript.**

*+ Figs 9, 10, 16: Could these be stacked as a three-panel figure? I felt the need to flip back and forth to compare the different versions of this figure.*
**We have stacked the original Figures 9 and 10 as a single figure, to form Figure 8 in the revised manuscript. We have kept the original Figure 16 (now Figure 13) separate, since a three-panel figure made each individual image too small and difficult to read.**

*+ Figs 12, 15: Could the lines in figure 15 be added as dashed lines in figure 12 if that's not too distracting?*
**Done; see Figure 10 in the revised manuscript.**

*Table 1: Could the cloud droplet number concentration be added to the table? For the run with predicted droplet concentration, a range of values could be given that could be different between the two times if appropriate.*
**Given that CASIM_NO_PROC is the only simulation where the droplet number concentration is predicted, we have decided not to add this information to Table 1. Note that the range of predicted droplet number concentrations is already quoted in the original manuscript (on page 8 line 14).**

*Using an observationally well-characterized case in southern West Africa, the role of sedimentation of cloud droplets in determining liquid water path and heights of low-level clouds, once established, is illustrated using large-eddy simulation and microphysical parameterizations with and without sedimentation. Controls by cloud drop number concentration (and drop size) on the extent to which sedimentation is effective in determining cloud height and water path are also discussed.*
*This is an important paper, extending earlier work on marine clouds and potential cloud-aerosol interactions related to sedimentation to land clouds. Although many questions remain, especially related to the roles of interactive surface fluxes of heat and moisture, which are not considered here, the paper advances knowledge of low-clouds in a region where they play an important role in the regional surface radiation balance and may be subject to strong aerosol interactions.*

*The paper is generally well written. While I agree with RC 1 about consolidating figures, the study offers the opportunity to illustrate some of the physical mechanisms at play in more detail, and I suggest the authors consider doing so. Specifically:*

*1. On Fig. 6, characterize the cloud base altitude for SIMPLE_CLOUD as has been done for CASIM_NO_PROC.*
**As suggested, we have now included timeseries plots from the new CASIM_NO_SED simulation and overlaid these on to Figure 5b in the revised manuscript.**

*2. A figure illustrating the different mixing ratio profiles for the cases in Table 1 would help to visualize the corresponding differences in sedimentation in these cases. A figure showing some measure of droplet size would also be helpful.*
**We have given careful consideration to the reviewer's suggestion of adding new figures, both here and in relation to comment #4 below. In light of the overall need to reduce / consolidate the overall number of figures in the manuscript, we have decided not to add additional figures to complement table 1. Instead we have introduced a new figure to quantify the rates of evaporative cooling and longwave radiative cooling, in response to comment #4 below.**

*3. What are the units of the field shown in Fig. 2?*
**There is no unit as such for the images from the cloud camera. The color scale is dependent on the emissivity of the sky and, consequently, the brightness temperature, where red colours indicate warm temperatures and blue cold temperatures. The images are used solely to provide a qualitative estimate of the horizontal homogeneity of the cloud deck (see also response to reviewer #1).**

*4. The importance of long-wave radiative cooling is discussed for three features of the simulation: (1) cloud formation and maintenance (p. 5, ll. 25-28; p. 7, ll. 15-18; p. 9, l. 25); (2) formation of stable layer near surface overnight (pp. 5-6); and (3) reduced long-wave cooling near cloud top due to sedimentation (p. 7, l. 9). Figures illustrating radiative cooling rates would illustrate these points effectively. Also, with sedimentation, both radiative and evaporative cooling are reduced near cloud top. A figure comparing these rates would be very useful in understanding the relative roles of the two processes.*

We have re-run simulations of CASIM_NO_PROC and SIMPLE_CLOUD (now replaced with CASIM_NO_SED as discussed in response to reviewer #1) to include additional profile diagnostics of the longwave cooling rate and evaporative cooling rates. We have added an additional figure to the manuscript to compare these rates in both simulations, as shown below:

[Figure]

*Figure 11. Domain average vertical profiles of a) longwave radiative heating rate (K hr −1 ) and b) condensation heating rate (K hr −1 ) calculated as temporal means between 0500-0530 UTC for CASIM_NO_PROC (blue) and CASIM_NO_SED (orange). Longwave and condensation heating rates for the period 0630-0700 UTC are shown in c) and d) respectively.*

The following text has also been added to the Results section of the revised manuscript:

"A closer inspection of Fig. 11 reveals more information about the relative roles of radiative cooling and evaporative  cooling in the evolution of the cloud layer. In both simulations, it is clear that radiative cooling is the dominant process, with peak rates that are typically an order of magnitude larger than those produced by evaporation near cloud top. The absence of sedimentation in CASIM_NO_SED results in larger cooling rates associated with both processes. However, the increase in longwave cooling rates is relatively modest - around 37% by 0700 UTC - whereas evaporative cooling rates increase by a factor of 2 within the same time period. Thus in relative terms, the effect of sedimentation appears to have the largest impact on rates of evaporative cooling."

---

## Author Response (AR2)

The authors would like to thank the Co-Editor for approving our changes to the manuscript. As suggested, the labels in Figures 8 and 13 have been updated, and a label has been added to Figure 14 for consistency. These changes are included within the revised manuscript.

Chris Dearden.